# Vertical mixing alleviates autumnal oxygen deficiency in the central North Sea

Charlotte A. J. Williams[1], Tom Hull[2], Jan Kaiser[3], Claire Mahaffey[4], Naomi Greenwood[2], Matthew Toberman[5] and Matthew R. Palmer[6]

[1]National Oceanography Centre, Liverpool, L3 5DA, UK
[2]Centre for Environment, Fisheries and Aquaculture Science, Lowestoft, NR33 0HT, UK
[3] University of East Anglia, Norwich, NR4 7TJ, UK
[4]Department of Earth, Ocean and Ecological Sciences, University of Liverpool, L69 3GP, UK
[5]Scottish Association for Marine Science, Oban, Scotland, PA37 1QA, UK
[6]Plymouth Marine Laboratory, Plymouth, PL1 3DH, UK

*Correspondence to*: Charlotte A. J. Williams (chwill@noc.ac.uk)

**Abstract.** There is an immediate need to better understand and monitor shelf sea dissolved oxygen ($O_2$) concentrations. Here we use high-resolution glider observations of turbulence and $O_2$ concentrations to directly estimate the vertical $O_2$ flux into the bottom mixed layer (BML) immediately before the autumn breakdown of stratification in a seasonally stratified shelf sea. We present a novel method to resolve the oxycline across sharp gradients due to slow optode response time and optode positioning in a flow "shadow zone" on Slocum gliders. The vertical $O_2$ flux to the low-$O_2$ BML was found to be between 2.5 to 6.4 mmol $m^{-2}$ $d^{-1}$. Episodic intense mixing events were responsible for the majority (up to 90 %) of this oxygen supply despite making up 40 % of the observations. Without these intense mixing events, BML $O_2$ concentrations would approach ecologically concerning levels by the end of the stratified period. Understanding the driving forces behind episodic mixing and how these may change under future climate scenarios and renewable energy infrastructure is key for monitoring shelf sea health.

## 1. Introduction

The global ocean is losing oxygen (Gilbert et al., 2010; Schmidtko et al., 2017), and the risk of deoxygenation in the productive coastal and shelf sea regions is increasing (Wakelin et al. 2020, Mahaffey et al., 2023). Oxygen decline and deficiency are already evident in several shelf seas (Grantham et al., 2004; Diaz & Rosenberg, 2008; Gilbert et al., 2010; Greenwood et al., 2010; Queste et al., 2013, Mahaffey et al., 2023) with recent model studies (NEMO-ERSEM, Butenschön et al., 2016) estimating that large regions of the Northwest European continental shelf seas (325 000 to 400 000 km²) have the potential to become seasonally deficient in dissolved oxygen ($O_2$) in late summer (Ciavatta et al., 2016) and in future climate scenarios (Wakelin et al., 2020). Although hypoxia (defined as $O_2$ mass concentrations below 2 mg $L^{-1}$) is never reached in UK shelf sea waters, $O_2$ deficiency ($< 6$ mg $L^{-1}$; OSPAR, 2013) is still lethal for some fish and molluscs (Vanquer-Sunyer & Duarte, 2008). The shelf seas around the UK are worth an approximate £47 billion per year (in terms of gross value added (GVA); Foresight Future of the Sea Report, 2018), with commercial fishing being a vital contributor. The increasing threat of oxygen deficiency could have serious economical as well as environmental impacts.

In shelf seas, the surface mixed layer (SML) is well-oxygenated due to air-sea gas exchange and net biological production of oxygen. However, the shelf sea thermocline acts as a physical barrier between the oxygenated, nutrient-deplete productive SML and the dark, nutrient-rich bottom mixed layer (BML) (Sharples et al., 2001, Mahaffey et al., 2023). In the

BML, net $O_2$ removal can occur as a result of restricted ventilation due to seasonal thermal stratification, oxygen consumption via pelagic and benthic respiration of organic matter and nitrification. An important mechanism in sustaining high productivity in shelf seas is the diapycnal upward mixing of nutrients across the base of the thermocline barrier (Sharples et al., 2007; Rippeth, 2005; Rippeth et al., 2014; Williams et al., 2013a; Davis et al., 2014; Brandt et al., 2015).

There is also potential for diapycnal mixing to help alleviate $O_2$ deficiency in the BML by providing a downward turbulent flux of $O_2$ across the thermocline (Queste et al., 2016; Rovelli et al., 2016; Williams et al., 2022). Diapycnal mixing is driven by the barotropic tide, internal waves and wind-driven inertial oscillations (Burchard and Rippeth, 2009; Sharples et al., 2007; Inall et al., 2000; Williams et al., 2013b), with the fluxes of nutrients and oxygen across the thermocline being dominated by episodic mixing events driven by these processes (Sharples et al., 2007; Williams et al., 2013a; 2013b).

Future shelf sea scenarios indicate earlier onset of stratification, prolonged periods of stratification and stronger thermocline stability (Holt et al., 2022; Sharples et al., 2013; Lowe et al., 2009; Meire et al., 2013), all of which will affect BML ventilation, $O_2$ dynamics and ecosystem health (Wakelin et al., 2020). To enable effective water quality and ecosystem management, as well as to assess the impact of the changing climate and increasing renewable energy infrastructure on our shelf seas, it is imperative to gather sufficient $O_2$-related data at high spatio-temporal resolution alongside hydrographic

measurements to determine trends in dissolved $O_2$ concentrations in our shelf seas. However, $O_2$ concentration measurements are seasonally and/or spatially limited in vast regions of UK shelf seas (Große et al., 2016, Mahaffey et al. 2023). The use of oxygen optodes on marine autonomous vehicles, including gliders, has greatly amplified the number of oceanic $O_2$ measurements in coastal and shelf seas available for ecosystem assessment (Mahaffey et al., 2023; Williams et al., 2022; Queste et al., 2016). Gliders can also quantify oceanic turbulence with microstructure shear probes.

In this study, we present glider-based high-resolution $O_2$ concentration and turbulence dissipation rate measurements immediately before the autumn breakdown of stratification in the North Sea. These measurements were carried out as part of the AlterEco ("An Alternative Framework to Assess Marine Ecosystem Functioning in Shelf Seas") project, which relied on gliders alone to monitor ecosystem functioning continuously over 17 months. Using this dataset, we were able to quantify and assess the diapycnal $O_2$ flux between SML and BML using co-located measurements of both $O_2$

concentration, $C(O_2)$, and turbulence dissipation rate, and demonstrate the importance of diapycnal mixing in the alleviation of oxygen deficiency at the end of the stratified period in autumn.

## 2. Materials and Methods

### 2.1 Glider specifications and deployment

The AlterEco project was a 17-month long continuous glider sampling campaign in the seasonally stratified North

Sea. One of the aims of AlterEco was to demonstrate the feasibility of monitoring the health of the marine ecosystem using only gliders deployed from small boats within 50 miles of the coast. Once deployed, gliders were piloted to the sampling site in the Central North Sea (Fig. 1). A single Slocum glider (Teledyne Webb Research, Falmouth, USA) from the Marine Autonomous and Robotic Systems group at the National Oceanography Centre (unit 444, called "Kelvin") was deployed during late autumn (September to November 2018) to measure and quantify the vertical oxygen flux just before the

breakdown of stratification. Glider "Kelvin" was deployed on 28 September 2018 from RV Princess Royal at 55.38° N, 0.51° W, and took approximately 9 days to reach the AlterEco study site at 56.2° N, 2° E (Fig. 1). Here, glider "Kelvin" was instructed to conduct virtual mooring dives from 18 October until 9 November (22 days), and then piloted back near shore for recovery on 2 December. A 'virtual mooring' is a glider flight pattern where the glider does not head towards a new waypoint, it attempts to hold a geographic position by turning back to its origin after each dive.

Glider "Kelvin" was equipped with a MicroRider microstructure package (Rockland Scientific International) to measure turbulent shear stress (see Palmer et al., 2015 for full details), a Seabird SBE42 CTD sensor to measure temperature, salinity, and pressure, and an Aanderaa 4831 oxygen optode to measure $C(O_2)$. Measurements were taken within 15 m of the bed and 5 m of the surface on most dives, with each dive taking about 20 minutes. The reason for cautious piloting away from the seabed was to protect the delicate probes on the MicroRider in an area known to host many

banks and changing bathymetry.

Glider salinity data were corrected for thermal inertia following Palmer et al. (2015). The glider AA4831 optode displays relatively long lag times (of the order of tens of seconds) and the positioning of the oxygen optode in a flow "shadow zone" at the back of the Slocum glider is not ideal either for resolving sharp oxyclines (Nicholson et al., 2008; Moat

et al., 2016). Oxygen concentrations were corrected for optode membrane lag following Bittig et al. (2014). However, where
the oxycline was too steep and full $C(O_2)$ variations could not be resolved, the data were omitted. Occasionally, the glider
turned close to the deep pycnocline (see Fig. 2b). This meant that the glider was not sampling for long enough to resolve the
$C(O_2)$ minimum in the narrow bottom mixed layer (BML) before it turned to climb. This resulted in unresolved BML $C(O_2)$.
Glider optode oxygen were calibrated against Winkler analysed discrete samples (surface and bottom triplicate, Fig. A1).
The AA4831 optode has been shown to have low drift ($< 0.5$ % $a^{-1}$) and good precision (better than 0.2 mmol $m^{-3}$)
(Körtzinger et al., 2004; Nicholson et al., 2008; Johnson et al., 2010; Champenois & Borges, 2012). Optode drift was
estimated to have been 0.004 % $d^{-1}$ in this study by comparing all optode drift rates throughout the AlterEco project.

## 2.2    Oxygen flux calculations

Following Sharples et al. (2007), the oxygen flux, $J(O_2)$, was estimated as

$$J(O_2) = m \frac{\Gamma \epsilon \rho}{g}$$  [1]

where g = 9.81 m $s^{-2}$, $\rho$ is water density (in kg $m^{-3}$), $\varepsilon$ is the turbulent kinetic energy dissipation rate (in W $kg^{-1}$) and m =
$dC(O_2)/d\rho$ (in mmol $kg^{-1}$) is the oxygen concentration gradient with respect to density. The unit of $J$ is mmol $m^{-2}$ $s^{-1}$. The
dimensionless mixing efficiency $\Gamma$ is defined as the ratio of potential energy gained relative to energy used to activate mixing
and assumed to be constant at 0.2 for the stratified water column (Osborn, 1980; Tweddle et al., 2013; Williams et al.,
2013a; 2022). While there is ongoing discussion on the assumption of a constant mixing efficiency in stratified fluids, no
study has arrived yet at a conclusive improvement, so this simple solution has been employed here as current best practice
(Gregg et al. 2018).

As previously discussed, the optode membrane lag combined with the positioning of the optode meant that we
could not resolve the oxycline accurately. This issue is particularly acute for shelf seas where there are large $C(O_2)$ and
temperature gradients. To overcome this problem, we have substituted m with the product of the CTD temperature–density
gradient, $dT_{CTD}/d\rho$, (approximately linear over a sufficiently narrow temperature range) and the optode oxygen
concentration–optode temperature gradient, $dC(O_2)/dT_{opt}$. This gives the following revised version of Eq. [1]:

$$J(O_2) = \frac{dT_{CTD}}{d\rho} \frac{dC(O_2)}{dT_{opt}} \frac{\Gamma \epsilon \rho_{CTD}}{g}$$  [2]

Testing this method on an independent dataset in a nearby region which had both accurate measurements of the oxygen-
density gradient, and a slow response AA4831 optode to compare against, indicated that our method calculates the oxygen-
density gradient to within 13 % of the true oxygen-density gradient value (Appendix). The base of the pycnocline defines the
boundary between the SML and BML and represents the interface for diapycnal mixing and $O_2$ fluxes. In the BML, tidal
mixing vertically homogenizes density and C(O2). The base of the pycnocline was defined as the depth that marked the top
of the BML (Fig. 2b). Confidence limits (95 %) for $J(O_2)$ were calculated using Efron Gong bootstrap resampling method
(Efron and Gong, 1983).

## 3.    Results

### 3.1  Water column characteristics

Glider "Kelvin" measured well-defined late autumn stratification in the water column with a depth of 85 m and a
relatively deep (50-60 m) thermocline both on transit to and once on station (Figure 2b). Stratification was thermally driven,
with the 50-60 m thick SML at 11.5 °C being 4.5 °C warmer than the comparably thinner 20-30 m bottom mixed later at 7.0
°C (Figure 2a). The SML temperature decreased by 2.3 °C to 9.2 °C over the 22 day-sampling period. There were further
layers of vertical stratification within the SML and above the BML at the beginning of the deployment until the 23 October
(Figure 2b). The relatively deep pycnocline was observed until early November, later than predicted by the regional 7 km-

resolution Atlantic Margin Model AMM7 (Fig. C1). The glider started travelling west on 9 November to the recovery site,
but it is unclear whether the glider moved into colder water or was capturing additional heat loss from the SML (Fig. 2).

The turbulent kinetic energy (TKE) dissipation rate $\varepsilon$ within the water column ranged between $1\times10^{-10}$ and $1\times10^{-6}$ W kg$^{-1}$, with the strongest mixing observed at the surface and the lowest $\varepsilon$ values observed within the mid-water below the boundary of surface mixing. Prolonged periods of intense wind-driven boundary mixing ($\varepsilon = 1\times10^{-6}$ W kg$^{-1}$) at the surface were evident from the measurements of TKE dissipation, with this turbulence occasionally penetrating and eroding the top of
the pycnocline, as observed between 23 and 27 October (Fig 2c). In comparison to the surrounding midwater $\varepsilon$ levels ($1\times10^{-9}$ W kg$^{-1}$), enhanced $\varepsilon$ was observed within the pycnocline when the glider dived deep enough to sample across it (Fig 3c). Furthermore, episodic intense bursts of $\varepsilon$ ($1\times10^{-6}$ W kg$^{-1}$) were also occasionally observed within the pycnocline (Fig. 2c and Fig. 3c). However, the glider did not measure clear bottom boundary turbulence driven by the barotropic tide due to having to turn within 15m of the bed and thus not always being in the narrow BML layer as discussed.

The deep pycnocline separated the oxygen-replete SML (280 µmol kg$^{-1}$, 100 % saturation) from the oxygen-depleted BML (240-250 µmol kg$^{-1}$, 75 % saturation; Fig. 3). Evidently the glider only sampled across the pycnocline and into the BML on two separate occasions (Fig. 2). Therefore, for this study the two periods are treated as two separate time series (A, from 18 to 30 October 2018; B, from 8 to 13 November 2018 - travelling SW back to shore for recovery from 9 November) for flux calculations.

**3.2 Diapycnal mixing of O₂**

The instantaneous diapycnal turbulent kinetic dissipation rate $\varepsilon$ over both time series A and B ranged from $1\times10^{-9}$ to $1\times10^{-7}$ W kg$^{-1}$, with the average for time series A calculated as $\varepsilon_{pycno} = (8.3\pm2.0)\times10^{-8}$ W kg$^{-1}$ (95 % confidence interval). For time series B, $\varepsilon_{pycno}$ was slightly lower at $(2\pm0.8)\times10^{-8}$ W kg$^{-1}$ (Table 1). The instantaneous $J(O_2)$ flux into the BML ranged between $10^{-6}$ and $10^{-4}$ mmol m$^{-2}$ s$^{-1}$ (Fig. 3d) with the highest values corresponding to the highest $\varepsilon_{pycno}$ values (Fig.
3c). Mean $J(O_2)$ was $(5.4\pm1.0)$ mmol m$^{-2}$ d$^{-1}$ and $(3.5\pm1.0)$ mmol m$^{-2}$ d$^{-1}$ for time series A and B, respectively (Table 1). The mean $J(O_2)$ was higher in time series A due to a significantly larger mean $\varepsilon_{pycno}$ (Table 1), despite experiencing a weaker oxygen density gradient compared to time series B (-34.4 and –92.5 respectively). The stronger oxygen gradient in time series B was due to the SML decreasing in temperature between time series A and time series B (Fig. 2b), resulting in a lower density gradient between the SML and BML, which in turn increased the observed oxygen-density gradient.

There was a total of 504 measurements of $\varepsilon_{pycno}$ over both time series A and B and of these measurements there were 204 episodic mixing events or 'spikes' (defined as periods when $\varepsilon_{pycno} > 10^{-7.5}$ W kg$^{-1}$, which was 25 standard deviations from the mean) which made up 40 % of the observations of $\varepsilon_{pycno}$. Despite episodic events making up such a small proportion of the $\varepsilon_{pycno}$ observations, when these were omitted, the recalculated mean $\varepsilon_{pycno}$ decreased from $(8.3\pm2.0)$ to $(1.5\pm0.1) \times 10^{-8}$ W kg$^{-1}$ for time series A, and from $(2.0\pm0.8)$ to $(0.7\pm0.2)\times10^{-8}$ W kg$^{-1}$ for time series B. We used these mean $\varepsilon_{pycno}$ values
(with spikes omitted) to recalculate $J(O_2)$ and found it decreased to 1.0 mmol m$^{-2}$ s$^{-1}$ for time series A and time series B. This equated to episodic mixing events making up 74 to 89 % of the observed $J(O_2)$ into the BML.

**4. Discussion**

Diapycnal mixing may act to alleviate oxygen deficiency in shelf seas by providing oxygen to the BML during the earlier periods of spring and summer stratification (e.g., Rovelli et al., 2016; Williams et al., 2022), but direct measurements
of the diapycnal oxygen flux in shelf seas are sparse both temporally and spatially. In this study, we present the first estimates of autumn diapycnal oxygen fluxes in a temperate shelf sea immediately before the breakdown in stratification, thus at the period where the BML had spent the longest time isolated from the atmosphere and O₂ concentrations were likely to be at their lowest. We estimated the oxygen flux using co-located measurements of both $C(O_2)$ and TKE dissipation rate on an autonomous underwater glider. Furthermore, we have presented a numerical solution to resolve oxygen fluxes where
there are limited measurements within the oxycline and in the thin BML. Our method of using the optode oxygen concentration–optode temperature gradient can prove very useful where sharp gradients across the pycnocline are unable to be resolved by the glider.

Diapycnal mixing supplied $O_2$ fluxes between 2.5 and 6.4 mmol $m^{-2}$ $d^{-1}$ (lower and upper end of the 95 % confidence intervals) into the relatively thin (20 to 30 m) BML in autumn. This is the first time a thin BML so late in autumn has been observed in the Northern North Sea, indicating that wind and tide were unable to erode the pycnocline in this relatively energetic region. Integrated over the autumn period (120 to 150 days), such an oxygen flux would equate between 0.3 and 1.0 mol $m^{-2}$ of oxygen being supplied across the pycnocline into the BML via diapycnal mixing. Assuming a 20 to 30 m thick BML as typical for this time of year in this area, this equates to a volumetric increase in oxygen content of 10 to 50 µmol $kg^{-1}$. At the end of the stratified period, the BML $O_2$ content was between 240 and 250 µmol $kg^{-1}$. Without the diapycnal supply of $O_2$, it would have been between 190 and 240 µmol $kg^{-1}$ (6.2 to 7.9 mg $L^{-1}$), the lower end of which approaches the ecologically concerning level of 6 mg $L^{-1}$. Rovelli et al. (2016) also measured diapycnal oxygen fluxes in the central North Sea (56.49° N, 2.98° E) during late summer 2009, and found that thermocline mixing provided the BML with a significant source of oxygen via thermocline mixing (18 to 74 mmol $m^{-2}$ $d^{-1}$). These notably large summer fluxes of $O_2$ into the higher volume BML than shown in our study are shown by the authors to significantly reduce the depletion of oxygen that occurs as a result of biological uptake in the sediment. It is apparent then that diapycnal fluxes that here are shown to be mostly driven by episodic events, provide a vital mechanism for alleviating oxygen deficiency in the North Sea towards the end of the stratified period.

The supply of oxygen across the pycnocline has recently been found to alleviate oxygen deficiency in other regions, too. For example, observations using a moored profiler in the northwestern boundary current region of Japan/East Sea over a 6-month period were found to provide a downward oxygen flux of 0.3 mmol $m^{-2}$ $d^{-1}$ from the East Sea Intermediate Water (Ostrovskii et al., 2021). In the open ocean oxygen minimum zone (OMZ) off northwest Africa the diapycnal oxygen flux was found to contribute approximately one third (0.4 mmol $m^{-2}$ $d^{-1}$) of the biological demand of the OMZ core (Fischer et al., 2013). Evidently, vertical mixing may act to ventilate deeper waters across many different oceanic regions. However, in the shelf sea, this mechanism can also supply the deeper BML with fresh organic material from the euphotic zone to be remineralised (Garcia-Martin et al., 2018), thus contributing to biological oxygen consumption. The relationship between vertical mixing and oxygen concentrations below the thermocline is therefore more complex than being solely a ventilation mechanism.

## 5.    Conclusions

Glider oxygen concentration measurements using optodes with a relatively slow response to estimate fluxes in environments with sharp $O_2$ gradients present a real challenge due to optode membrane lag and optode positioning. Irrespective of this, such datasets considerably increase the spatio-temporal measurement resolution for a better assessment and understanding of the ocean dynamics in these regions, where ecological and marine management issues are of prime importance. Furthermore, in this study we have provided a solution to accurately calculate vertical fluxes using gliders, and have demonstrated how autonomous underwater vehicles are a low-cost and effective tool to potentially contribute to providing monitoring capability.

Coastal and shelf seas are increasingly being looked to as an energy source with the development of wind and tidal renewable energy, with the aim for 40 GW of energy being harnessed from the wind by 2030. Research on the impact of these floating and static wind farms on stratification, mixing and deoxygenation is premature, but a recent study has indicated that deoxygenation will increase in areas of the North Sea already at risk of low oxygen levels as a result of the presence of static wind farms increasing primary production but also reducing advective currents and bottom shear stress (Daewel et al., 2022). It is therefore imperative that we continue to monitor dissolved oxygen concentrations alongside measurements of hydrography and mixing in coastal and shelf sea waters, especially as offshore installations expand, with unknown consequences on ecosystem health. Furthermore, resolving the oxycline effectively in these environments is crucial to understand how potentially oxygen deficient bottom waters might evolve with varying mixing regimes in the future.

## 6.    Appendices

**Appendix A: Optode oxygen concentration calibration using discrete Winkler samples**

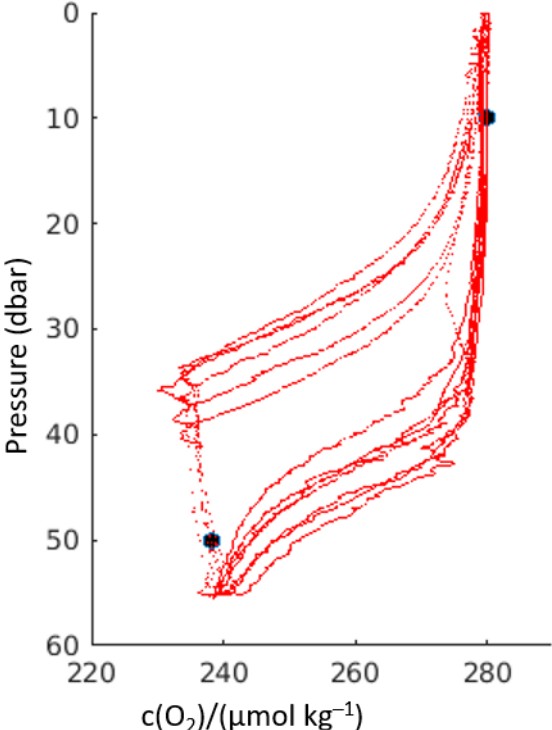

Figure A1: Vertical oxygen content profiles from the initial 6 glider dive and climbs at deployment of glider "Kelvin" (red lines). Black circles are the oxygen values from Winkler-analysed discrete samples taken in triplicate within the SML and BML. Note the severe optode membrane lag evident due to the disparity between dive and climb profiles.

## Appendix B: Comparison between oxygen concentration-density gradients measured directly using high-resolution optode data and reconstructed using Eq. 2

As we cannot accurately resolve the oxygen concentration gradient directly in our glider study due to only having a slow-response optode, our calculated $m = dT_{CTD} / d\rho * dC(O_2)/dT_{opt}$ cannot be directly compared to $m = dC(O_2)/d\rho$ at the time of survey reported on in the present manuscript. However, data from the same area of the North Sea at a different time of year (during late summer in the presence of a stronger vertical oxygen concentration gradient), when we did have both high-resolution oxygen concentration data (calibrated fast-response Rinko optode) and a lagged slow-response optode (Aanderaa 4831) for comparison (Fig. B3), show the validity of our approach to analyse the slow-response optode data.

Specifically, we have calculated the oxygen concentration-density gradient at the base of the pycnocline from the Rinko data (Fig. B1).

We have then used the glider CTD temperature $T_{CTD}$, lagged optode oxygen concentration $C(O_2;$ lagged), and the lagged optode thermistor temperature $T_{opt}$ at the base of the pycnocline (where 1027.5 kg m$^{-3}$ < $\rho$ < 1027.75 kg m$^{-3}$; Fig. B2) to calculate the first two terms of Eq. 2 and compare it with the high-resolution oxygen concentration-density gradient. The latter gradient obtained from linear regression of the data in Fig. B4 is $m = -168$ mmol kg$^{-1}$.

235         For comparison, the oxygen concentration gradient calculated using Eq. 2 is

$$m = dT_{CTD}/d\rho \ dC(O_2; \text{lagged})/dT(\text{thermistor}) = \text{-4.2 K m}^3\text{/kg} * 45 \text{ mmol m}^{-3}\text{/K} = -189 \text{ mmol kg}^{-1}$$

        This estimate is within 12.5% of the high-resolution value. This demonstrates that our method can be used to calculate the oxygen gradient reasonably accurately.

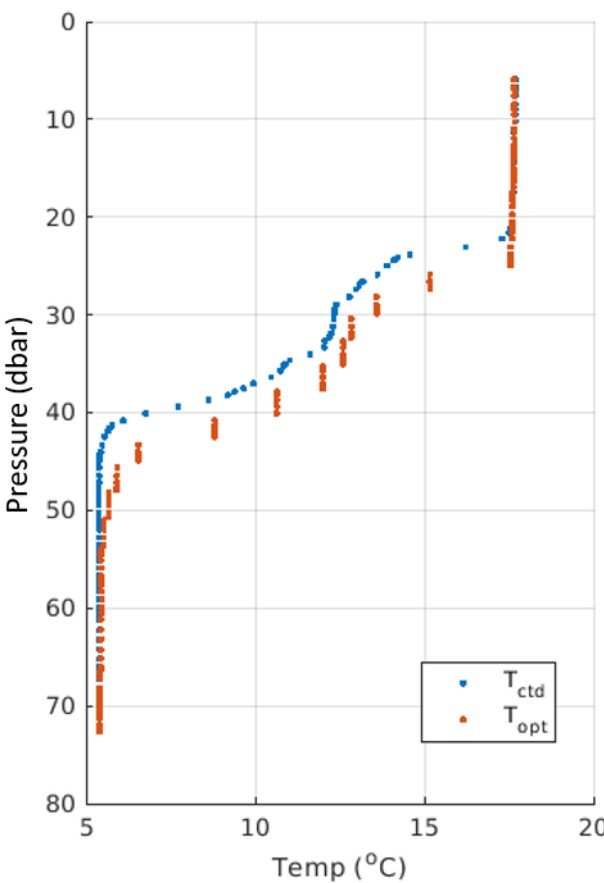

Figure B1: Vertical profile of CTD (blue) and optode thermistor temperature (red).

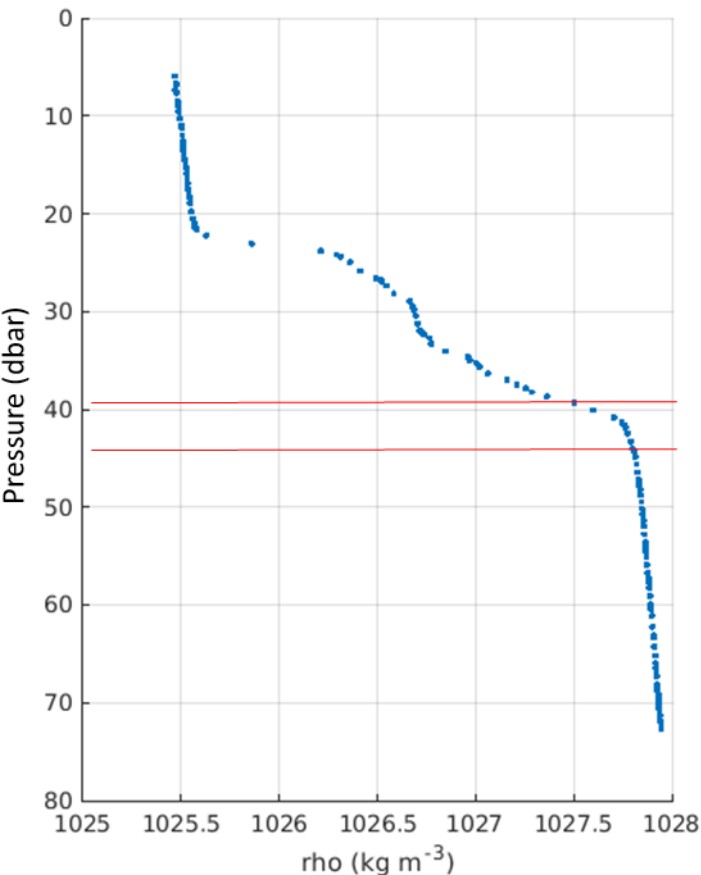

Figure B2: Density profile from the CTD showing two step structure, red lines mark the area where the oxygen concentration-density gradient has been calculated (1027.5 to 1027.75 kg m$^{-3}$).

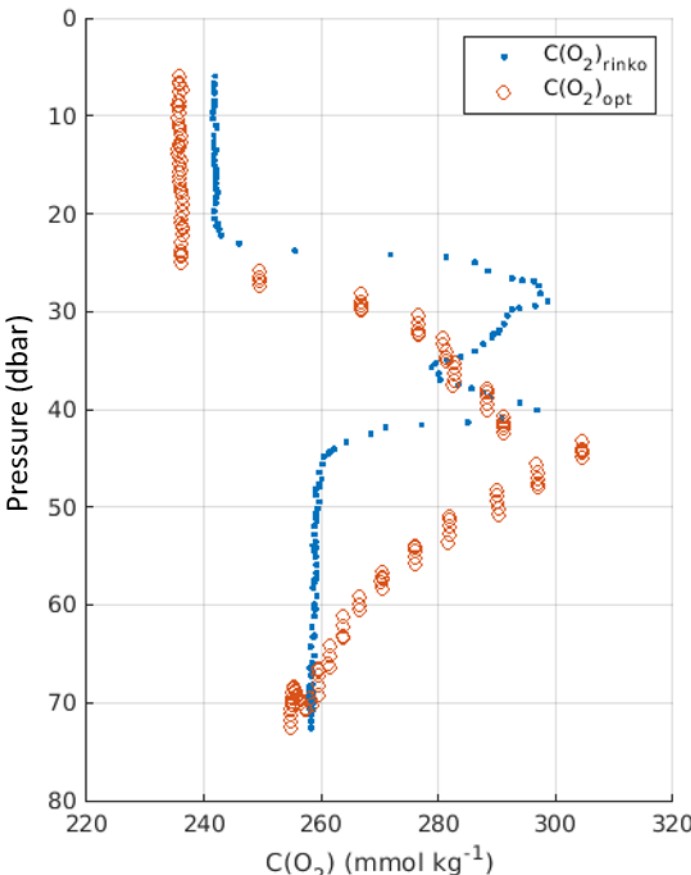

Figure B3: Fast response oxygen from RINKO (blue) and lagged Aanderaa optode (red), positioned like the optode in our glider study.

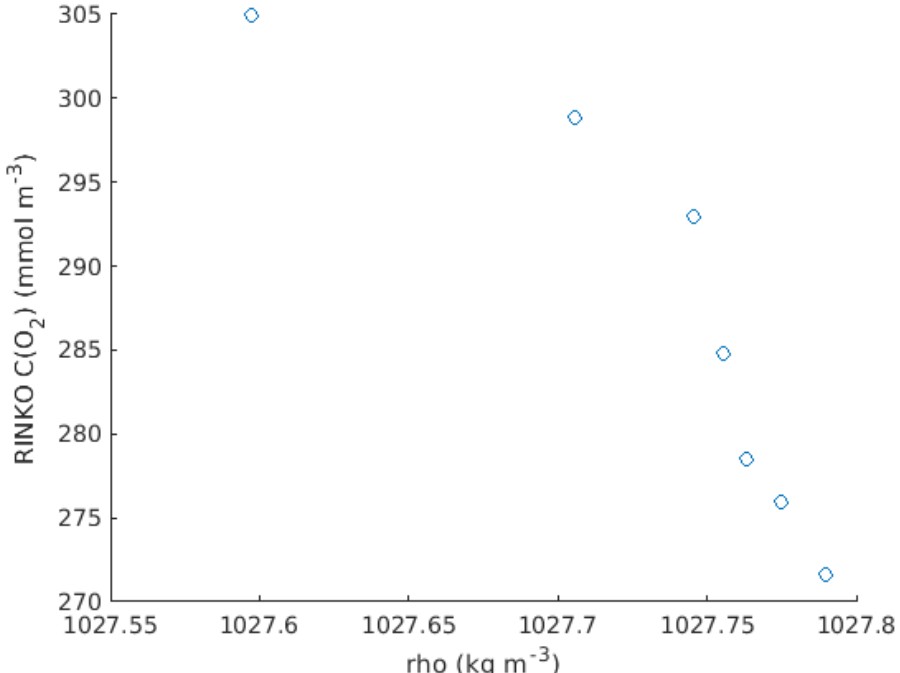

Figure B4: Oxygen concentration-density relationship from the high-resolution Rinko optode.

**Appendix C: Regional 7 km-resolution Atlantic Margin Model AMM7 temperature**

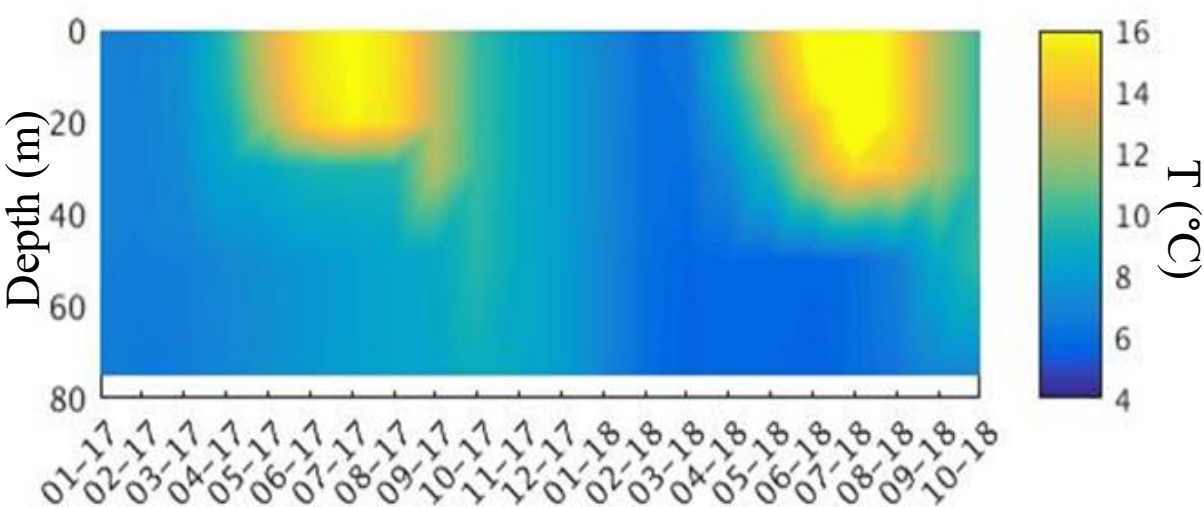

Figure C1: Regional 7 km-resolution Atlantic Margin Model AMM7 temperature for 2017 and 2018 from the same position as glider Kelvin (56.2° N, 2° E). AMM7 shows complete breakdown of stratification by October 2018, weeks earlier than observed.

**Data availability statement**

All data are publicly available via zenodo (FAIR compliant) from the BODC glider data archive under AlterEco 5 "Kelvin" (https://gliders.bodc.ac.uk/inventory/glider-inventory/) and under the NERC Open Government License.

**Author contribution**

C. Williams, T. Hull, J. Kaiser and M. Palmer designed the sampling campaign with the gliders. J. Kaiser developed the
oxygen optode temperature method to calculate the oxygen concentration gradient effectively. M. Toberman and M. Palmer processed the turbulence data from the OMG. C. Williams processed the rest of the glider data with input from C. Mahaffey, T. Hull and N. Greenwood. T. Hull performed oxygen Winkler analysis to calibrate the glider oxygen optode measurements. C. Williams prepared the manuscript with contributions from all co-authors.

**Acknowledgements**

We thank the captains and crew of the RV Princess Royal for their help and support at sea and all the scientists involved in the three cruises. We would also like to thank the Marine Autonomous & Robotics Systems (MARS) facility (National Oceanographic Centre, Liverpool) for deployment, recovery and piloting of "Kelvin". We are grateful to the UK Natural Environment Research Council (NERC) for funding the research cruises via the AlterEco project that supported this work (NERC grant reference NE/P013864/1).

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

**Figures**

**Figure 1. Bathymetric map showing AlterEco study region in the North Sea (inset) and glider tracks (black lines) to AlterEco sampling site at 56.2° N, 2° E.**

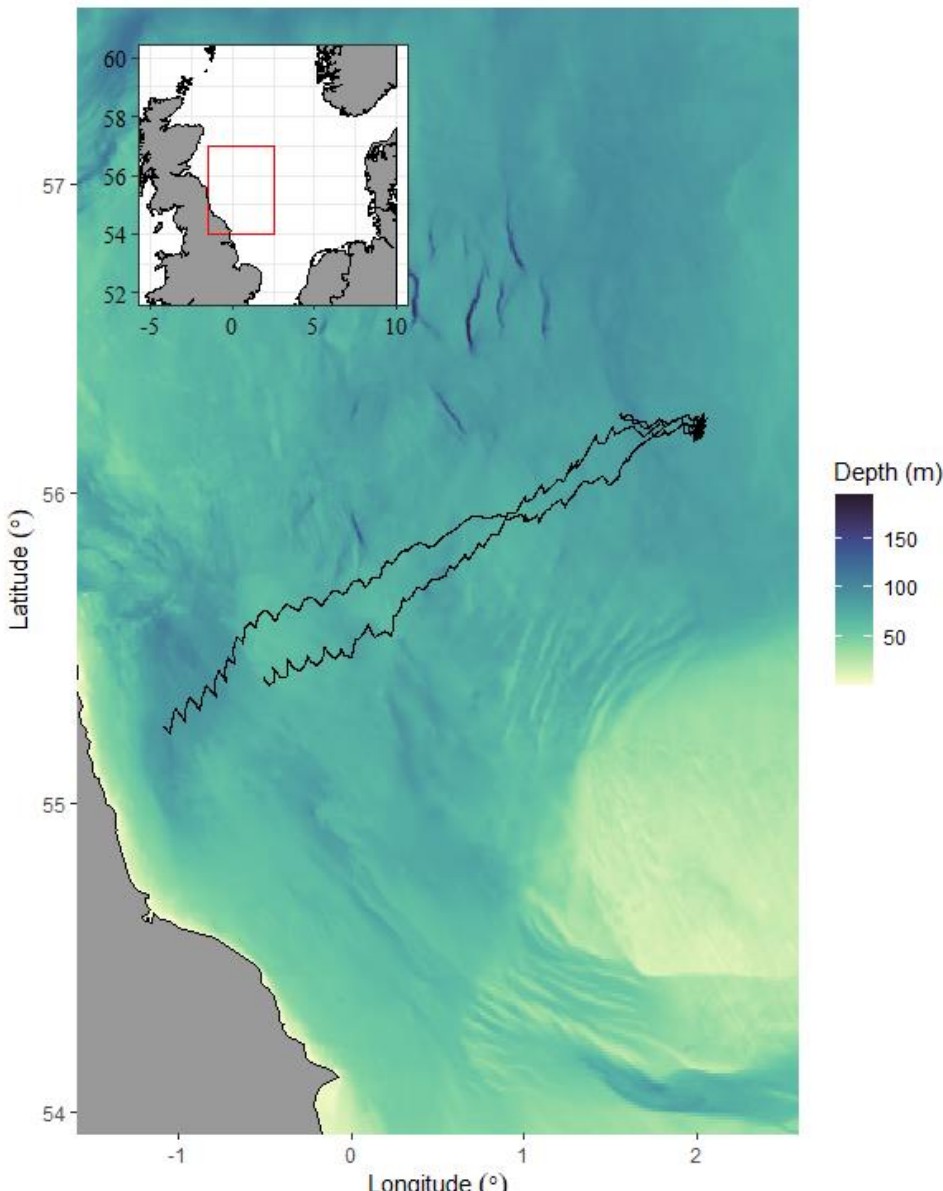


**Figure 2: Time series of a) glider position latitude and longitude, b) temperature *T*, and c) turbulent kinetic energy dissipation rate *ε*. The position of the pycnocline (thin black line) is shown in panel b. The seabed position derived from the glider altimeter is shown in panels b) and c).**

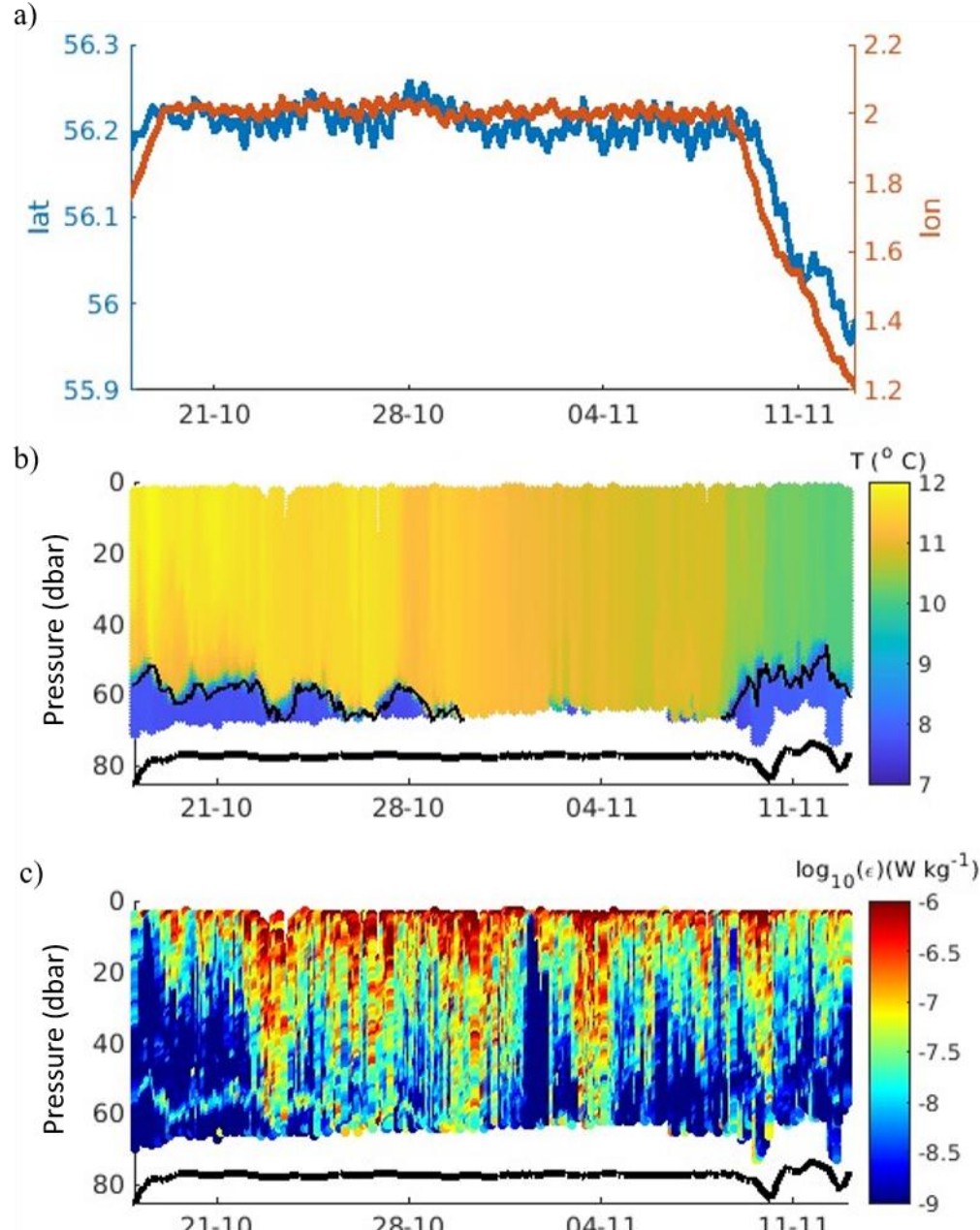


**Figure 3: Time series of a) oxygen content, b) oxygen saturation, c) turbulent kinetic energy dissipation rate $\varepsilon_{pycno}$ within the pycnocline (thin black line in panels a and b), and d) the instantaneous oxygen flux $J(O_2)$ across the pycnocline. The thick black line indicates the seabed position derived from the glider altimeter. Shaded grey boxes highlight where spikes in dissipation appear to coincide with enhanced oxygen fluxes.**

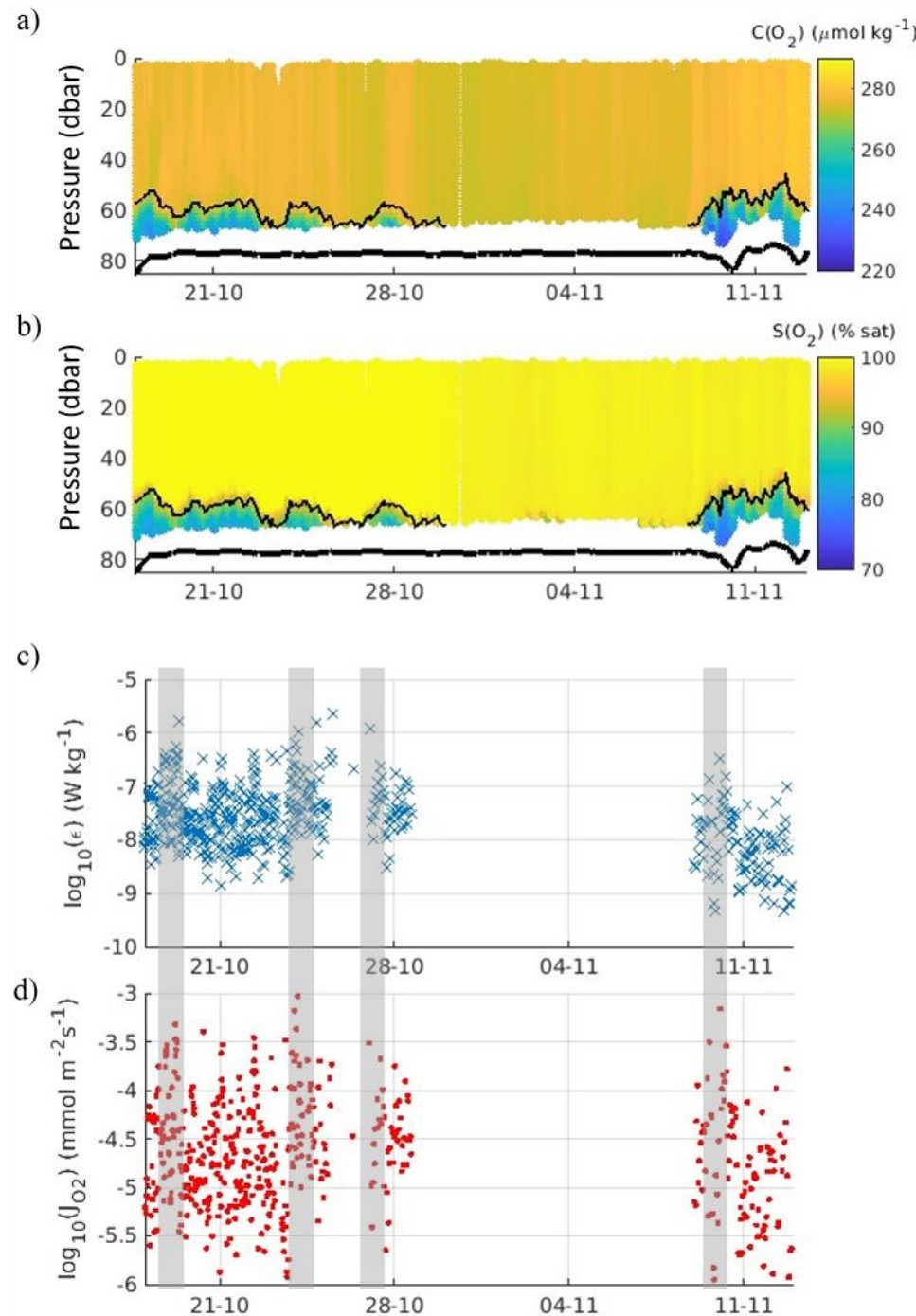


**Table 1: Estimates of pycnocline turbulent energy dissipation rate $\varepsilon_{pycno}$ and oxygen fluxes $J(O_2)$ during time series A and B, with and without episodic mixing events, defined as spikes where $\varepsilon > 10^{-7.5}$ W kg$^{-1}$. Uncertainties correspond to 95 % confidence intervals.**

| | $dC(O_2)/d\rho$ / (mmol kg$^{-1}$) | $\varepsilon_{pycno}$ / ($10^{-8}$ W kg$^{-1}$) | $J(O_2)$ / (mmol m$^{-2}$ d$^{-1}$) |
|---|---|---|---|
| Time series A, 18 to 30 October 2018 | −34.4 | 8.3±2.0 | 5.4±1.0 |
| Time series A, spikes removed | −34.4 | 1.5±0.1 | 1.0 |
| Time series B, 8 to 13 November 2018 | −92.5 | 2.0±0.8 | 3.5±1.0 |
| Time series B, spikes removed | −92.5 | 0.7±0.2 | 1.0 |
