# Peer review of "Vertical mixing alleviates autumnal oxygen deficiency in the central North Sea"

_Biogeosciences, 2023_

## Author Comment (AC1)

Response to reviewers

REVIEWER 1

The article explores the possibility of using gliders equipped with oxygen and turbulence sensors to estimate vertical oxygen fluxes, which has the potential to increase spatial and temporal resolution of measurements while reducing the carbon footprint. The approach of using the CTD values combined with the oxygen/temperature sensors seems to give good results and may be a good solution to the issue related with the slow response of oxygen sensors when using gliders in high gradient zones. I have a few comments:

In Methodology:

I'm missing information about sensor calibration/validation. For example, in the section Author contribution, Winkler analysis is mentioned. Maybe add a reference to this method section?

We thank the R1 for raising this important comment, the glider optode were calibrated against both Winkler analysed discrete samples and another calibrated glider optode. We agree this information needs to be within the manuscript and will add this both to the manuscript methodology text with further details of the calibration (calibration coefficients) within the appendix.

> Commented [to1]: something to put in the appendix?

o   Can the authors add a reference or explain what a 'virtual mooring dive' is?

Virtual mooring is a glider flight pattern where the glider does not head towards a new waypoint, it attempts to hold a geographic position by turning back to its origin after each dive. This will be added to the manuscript text.

In results:

o   Line 116: is it Fig 2a or Fig 2b?

This is a missed typo, apologies it should be Fig. 2b.

o   Line 125: Fig 2d doesn't exist

This should be 2c

o   Line 126: is it Fig 2c or 3c?

Both show enhanced mixing reaching the pycnocline, but Figure 3c only needs to be referred to (pycnocline TKE dissipation) so we will change this to Fig. 3c.

o   Line 146: please, add a reference to when considering a 'spike'

A spike is considered an event of magnitude 25 number of standard deviations from the mean.

o   Line 149: is there a typo at the beginning of the sentence?

We thank the reviewer for spotting this missed typo. It has been deleted.

In Conclusions:

o   The Conclusions section is too long and contains parts that really form a discussion. I would recommend adding a section for the discussion (which is missing in the article) and keeping the conclusion more concise. Conclusions should be about what the authors have found and not a discussion as found in the 3rd paragraph.

This is a comment also repeated by R2, and has been taken on board by the authors. A discussion will be included in the manuscript.

o   The final paragraph shouldn't be in the conclusions - it is too speculative to state that. The authors haven't provided any estimate of how gliders would be able to provide a monitoring tool for this specific case. Consider adding a section in the discussion and compare what is required with what the gliders can measure.

Citation: https://doi.org/10.5194/bg-2023-100-RC1

REVIEWER 2

RC2: 'Comment on bg-2023-100', Anonymous Referee #2, 07 Aug 2023  reply

General comments

The paper «  Vertical mixing alleviates autumnal oxygen deficiency in the central North Sea » by C. Williams and colleagues uses co-located oxygen and turbulence observations from a slocum glider to assess and understand the BML autumn oxygen supply on the North Sea Shelf.

As the authors explain, using oxygen observations from glider to estimate fluxes in sharp $O_2$ gradient is a real challenge (time lag, optode position ...). Nevertheless, such datasets considerably inscrease the spatio-temporal resolution of the measurements for a better assessment and understanding of the ocean dynamics in these regions, where ecological and economic issues are of prime importance.

Here, authors propose to combine CTD measurements with optode oxygen/temperature observations to improve the estimation of oxygen fluxes in areas with strong $O_2$ gradient. The results seem encouraging and well interpreted. Nevertheless, 2 key points driving the results are not sufficiently documented :

o   the m substitution : How (and How much) does m=dTctd / drho * dC(O2)/dTopt produce more accurate results than m=dC(O2)/drho ?

As we cannot accurately resolve oxygen gradient directly in our glider study due to only having a slow response optode, our calculated m=dTctd / drho * dC(O2)/dTopt cannot be directly compared to m=dC(O2)/drho. However, we have sourced some CTD measurements taken from the same area of the North Sea at a different time of year (during late summer and a stronger oxygen gradient), where we have reliable oxygen data (calibrated fast response Rinko optode) *and* a lagged slow response optode (Aanderaa 4831) for comparison as used in our manuscript.

Commented [WA2]: @tom.hull@cefas.gov.uk has comparison data I think from Dogger Bank?

We have calculated the oxygen density gradient from the base of the pycnocline from the RINKO data (see S.I. Fig. 1), this can be taken as our 'true' oxygen gradient.

We have then used the CTD temperature, lagged optode oxygen, and the lagged optode thermistor data at the base of the pycnocline (where 1027.5 <rho<1027.8) to calculate the first two terms of our equation 1 to compare our against the 'true' oxygen gradient.

The true oxygen gradient from linear regression of S.I. Figure 4:

m = 168 mmol/m4

The estimated oxygen gradient taken from (dT/drho) * (dC(O2_lagged)/dT_thermistor):

dT/drho = -4.2 deg C m^3/kg

DC(O2_lagged)/dT_thermistor = 45 mmol m-3/deg C

Giving dC(O2)/drho = -189 mmol/m4

Our estimated value of the oxygen gradient using lagged oxygen values from the slow response optode gives us an oxygen gradient within 12.5% of the true gradient value. Note that the slow response optode used in our example here has also not been calibrated (our glider slow response optode was calibrated), we would expect this to slightly change our value for dC(O2)/dT and be more accurate.

We think this represents independently that our method can be used to calculate the oxygen gradient pretty well, and if our manuscript is accepted we would add this method validation into our supplementary information (with more detail to be added on figures and calculations).

[Figure]

S.I. Figure 1: CTD profile of temperature (blue dots) with optode thermistor values (red).

[Figure]

S.I. Figure 2: Density profile from the CTD showing two step structure, red lines mark the area where the oxygen-density gradient has been calculated (1027.5 to 1027.75).

[Figure]

S.I. Figure 3: Fast response oxygen from RINKO (blue) and lagged and uncalibrated optode (red circles), like the optode used in our study on the glider.

[Figure]

S.I Figure 4: oxygen and density relationship from the RINKO (reliable oxygen measurement).

o   the spike determination. Add references, threshold definition/determination.  why $10^{-7.5}$ and not $10^{-6.5}$? 40 % of observations is not a « small proportion».

This is a fair point from the reviewer and one where we have not yet demonstrated our rationale towards. We selected this value based on being 25 standard deviations ($3.9 \times 10^{-8}$) from the mean TKE epsilon ($1.66 \times 10^{-8}$).

The addition of further information on these two points would add value to this new method and give weight to results.

We thank Reviewer 2 for these comments and agree that by addressing both of these points our manuscript would be improved. We would be happy to include this further information in the manuscript.

Specific comments

*Materials and Methods*

- L.98 : What is the potential impact of the simply solution employement (= constant mixing efficiency) on the results ?

Although there is no consensus on a mixing efficiency it is well documented that below a certain buoyancy Reynolds number (Reb) turbulence is not mixing diapycnally thus Eq (1) in the manuscript is valid only in developed turbulence. There are certainly limitations of this approach, especially in low energy systems as the Arctic, the Black Sea and the Baltic Sea. The low energy levels in this regions make it much more complicated to easily derive fluxes from single turbulence profiles (e.g. Holtermann et al. 2019, Scheifele et al. 2018).

The vast majority of studies have not conclusively arrived at a suitable improvement to that proposed by Osborn (1980), so in our manuscript we have employed this simple solution as also current best practice as outlined in Gregg et al., 2018.

- L101 to 105 : see 'm substitution' comments « general » section

As above, I think we have now provided adequate evidence that our method accurately resolves the oxycline.

- oxygen optode calibration/validation. Informations are missing in this section as it is mentioned in the author contribution.

This point was also raised by Reviewer 1 and we agree needs to be included in the manuscript. The glider optode was calibrated against both Winkler analysed discrete samples and another calibrated glider optode. We agree this information needs to be within the manuscript and will add this both to the manuscript methodology text with further details of the calibration (calibration coefficients) within the appendix.

*Results*

- L116 Figure 2b rather than 2a

Thank you this typo has been corrected.

- L125 : Figure 2d does not exist

This is a typo and should be 2c.

- L 127 : Fig 3c ?

Fig. 3c shows the episodic spikes so we have removed Fig. 2c from the figure referencing in L127

>> better referencing figures

- L.137: $10^{-7}$ ? not $10^{-6}$

This is a typo and has been corrected to $10^{-6}$

- L.139 : « with the highest values corresponding to the highest €pycno values » this is not visible on current figure 3.

We think Fig 3d mirrors 3c spikes quite clearly on Fig 3, but take reviewer 2's comment on board and are happy to mark the oxygen flux spikes out with grey boxes.

- L.142 to 144 : add figure references to help the reading

Noted, we have updated the manuscript to reference the figures more frequently.

- L.145 – 151 : see 'spike' comments « general » section

We have added a section here describing our rationale for chosing this value of epsilon yo describe a 'spike'.

*Conclusion*

Conclusion includes a discussion part that is missing in the paper. Change the section title or include a section « discussion »

Noted from R1 and R2, we have amended the manuscript to include a discussion section and a shorter conclusions section.

Technical corrections

- L. 57 - 61: add the time coverage of your dataset in this section compare to 17 month of data for Alter ECO for clarification. For example, <(Sep. to Nov.)> just before « autumn breakdown of stratification in the North Sea » and/or < Using this 3-month dataset >

Thank you have clarified.

- L70 : replace <('Kelvin', unit 444)> by <(called « Kelvin », unit 444)>

Thank you we have changed this.

- L75 : replace <Kelvin was equipped … > by <Glider « Kelvin » was equipped … > as line 113

- L118 : add model references.

> This is in comparison to AMM7 temperature (results not shown) we can add this to supplementary information if needed.

[Figure]

Plot shows temperatures (deg C) from AMM7 for the area where the glider was sampling, stratification shown to break down by October 2018.

- L142 : replace <-34-4> by <34.4>

Changed

- L150 1.0 mmol m-2 s-1 for A and 1 mmol m-2 s-1 for B > harmonize

Have changed thank you.

- figure 3  caption : add <(black line)> after <within the pycnocline>

Changed

- Explain TKE abbreviation in the text

Done

- Figure improvement suggestions : add bottom depth of the shelf sea on the sections of Figure 2 and 3 to complete them

We agree that this would improve the figures.

---

## Author Response (AR1)

**Response to reviewers (original comments: black; our response: red)**

**REVIEWER 1 ([https://doi.org/10.5194/bg-2023-100-RC1](https://doi.org/10.5194/bg-2023-100-RC1))**

The article explores the possibility of using gliders equipped with oxygen and turbulence sensors to estimate vertical oxygen fluxes, which has the potential to increase spatial and temporal resolution of measurements while reducing the carbon footprint. The approach of using the CTD values combined with the oxygen/temperature sensors seems to give good results and may be a good solution to the issue related with the slow response of oxygen sensors when using gliders in high gradient zones. I have a few comments:

I'm missing information about sensor calibration/validation. For example, in the section Author contribution, Winkler analysis is mentioned. Maybe add a reference to this method section?

We thank the R1 for raising this important comment, the glider optode were calibrated against Winkler analysed discrete samples in the surface and bottom mixed layers. We agree this information needs to be within the manuscript and have added this to the manuscript (L90) methodology text with further details of the calibration (calibration coefficients) in the Appendix.

o   Can the authors add a reference or explain what a 'virtual mooring dive' is?

A virtual mooring dive is a glider flight pattern where the glider does not head towards a new waypoint. Instead, it attempts to hold a geographic position by turning back to its origin after each dive. This has been added to the manuscript text (L75 – 77)

o   Line 116: is it Fig 2a or Fig 2b?

This is a missed typo, apologies; it has been corrected to Fig. 2b.

o   Line 125: Fig 2d doesn't exist

Corrected to 2c.

o   Line 126: is it Fig 2c or 3c?

Both show enhanced mixing reaching the pycnocline, but only Figure 3c needs to be referred to (pycnocline TKE dissipation) so this has been changed to Fig. 3c.

o   Line 146: please, add a reference to when considering a 'spike'

A spike is considered a value at least 25 standard deviations away from the mean. We have added this to the manuscript text.

o   Line 149: is there a typo at the beginning of the sentence?

We thank the reviewer for spotting this missed typo. It has been deleted.

o   The Conclusions section is too long and contains parts that really form a discussion. I would recommend adding a section for the discussion (which is missing in the article) and keeping the conclusion more concise. Conclusions should be about what the authors have found and not a discussion as found in the 3rd paragraph.

This is a comment also repeated by R2, and has been taken on board by the authors. A discussion has been included in the manuscript, and a shorter conclusions section.

o The final paragraph shouldn't be in the conclusions - it is too speculative to state that. The authors haven't provided any estimate of how gliders would be able to provide a monitoring tool for this specific case. Consider adding a section in the discussion and compare what is required with what the gliders can measure.

o We have amended the final paragraph to discuss how gliders might be able to contribute to the monitoring of coastal and shelf sea oxygen, together with what would need to be improved.

**REVIEWER 2 (https://doi.org/10.5194/bg-2023-100-RC2)**

The paper « Vertical mixing alleviates autumnal oxygen deficiency in the central North Sea » by C. Williams and colleagues uses co-located oxygen and turbulence observations from a slocum glider to assess and understand the BML autumn oxygen supply on the North Sea Shelf.

As the authors explain, using oxygen observations from glider to estimate fluxes in sharp $O_2$ gradient is a real challenge (time lag, optode position …). Nevertheless, such datasets considerably inscrease the spatio-temporal resolution of the measurements for a better assessment and understanding of the ocean dynamics in these regions, where ecological and economic issues are of prime importance.

Here, authors propose to combine CTD measurements with optode oxygen/temperature observations to improve the estimation of oxygen fluxes in areas with strong $O_2$ gradient. The results seem encouraging and well interpreted. Nevertheless, 2 key points driving the results are not sufficiently documented :

o  the m substitution : How (and How much) does m=dTctd / drho * dC(O2)/dTopt produce more accurate results than m=dC(O2)/drho ?

As we cannot accurately resolve the oxygen concentration gradient directly in our glider study due to only having a slow-response optode, our calculated $m$ = d$T_{CTD}$ / d$\rho$ * d$C(O_2)$/d$T_{opt}$ cannot be directly compared to $m$ = d$C(O_2)$/d$\rho$ at the time of survey reported on in the present manuscript. However, data from the same area of the North Sea at a different time of year (during late summer in the presence of a stronger vertical oxygen concentration gradient), when we did have both high-resolution oxygen concentration data (calibrated fast-response Rinko optode) and a lagged slow-response optode (Aanderaa 4831) for comparison (B3), show the validity of our approach to analyse the slow-response optode data, as we now explain in an Appendix.

Specifically, we have calculated the oxygen concentration-density gradient at the base of the pycnocline from the Rinko data (Fig. B1).

We have then used the glider CTD temperature $T_{CTD}$, lagged optode oxygen concentration $C(O_2;$ lagged), and the lagged optode thermistor temperature $T_{opt}$ at the base of the pycnocline (where 1027.5 kg m$^{-3}$ < $\rho$ < 1027.75 kg m$^{-3}$; Fig. B2) to calculate the first two terms of Eq. 2 and compare it with the high-resolution oxygen concentration-density gradient. The latter gradient obtained from linear regression of the data in Fig. B4 is $m$ = –168 mmol kg$^{-1}$.

For comparison, the oxygen concentration gradient calculated using Eq. 2 is

$m = \mathrm{d}T_{CTD}/\mathrm{d}\rho\ \mathrm{d}C(O_2;\ \mathrm{lagged})/\mathrm{d}T(\mathrm{thermistor}) = \text{-4.2 K m}^3/\text{kg} * 45\ \text{mmol m}^{-3}/\text{K} = -189\ \text{mmol kg}^{-1}$

This estimate is within 12.5% of the high-resolution value. This demonstrates that our method can be used to calculate the oxygen gradient reasonably accurately.

We have added this method validation as Appendix B, as well as modifying the main text (L112 – L114).

[Figure]

Figure B1: Vertical profile of CTD (blue) and optode thermistor temperature (red).

[Figure]

Figure B2: Density profile from the CTD showing two step structure, red lines mark the area where the oxygen concentration-density gradient has been calculated (1027.5 to 1027.75 kg m$^{-3}$).

[Figure]

Figure B3: Fast response oxygen from RINKO (blue) and lagged Aanderaa optode (red), positioned like the optode in our glider study.

[Figure]

Figure B4: oxygen concentration and density relationship from the high-resolution Rinko optode.

○ the spike determination. Add references, threshold definition/determination. why $10^{-7.5}$ and not $10^{-6.5}$? 40 % of observations is not a « small proportion».

This is a fair point where we have not yet clearly demonstrated our rationale. We selected this value based on being 25 standard deviations ($3.9 \times 10^{-8}$ W kg$^{-1}$) from the mean TKE epsilon ($1.66 \times 10^{-8}$ W kg$^{-1}$) and have added this reasoning to the manuscript text.

The addition of further information on these two points would add value to this new method and give weight to results.

We thank Reviewer 2 for these comments and have included this further information in the manuscript.

*Materials and Methods*

- L.98 : What is the potential impact of the simply solution employement (= constant mixing efficiency) on the results ?

Although there is no consensus on a mixing efficiency it is well documented that below a certain buoyancy Reynolds number ($Re_b$), turbulence is not mixing diapycnally; thus Eq (1) in the manuscript is valid only in developed turbulence. There are certainly limitations of this approach, especially in low-energy systems as the Arctic, the Black Sea and the Baltic Sea. The low energy levels in these regions make it much more complicated to easily derive fluxes from single turbulence profiles (e.g. Holtermann et al. 2019, Scheifele et al. 2018).

The vast majority of studies have not conclusively arrived at a suitable improvement to the fixed value proposed by Osborn (1980), so in our manuscript we have employed this simple solution, which is also current best practice as outlined by Gregg et al., 2018.

- L101 to 105 : see 'm substitution' comments « general » section

In the Appendix B, we have now provided evidence that our method accurately resolves the oxycline.

- oxygen optode calibration/validation. Informations are missing in this section as it is mentioned in the author contribution.

This point was also raised by Reviewer 1 and the missing information has been included in the revised manuscript. The glider optode was calibrated against Winkler-analysed discrete samples, which is explained in detail in the new Appendix.

*Results*

- L116 Figure 2b rather than 2a

Thank you; this typo has been corrected.

- L125 : Figure 2d does not exist

This is a typo and should be 2c. Corrected.

- L 127 : Fig 3c ?

Fig. 3c shows the episodic spikes so we have removed Fig. 2c from the figure referencing in L127

- L.137: $10^{-7}$ ? not $10^{-6}$

This is a typo and has been corrected to  $10^{-6}$

- L.139 : « with the highest values corresponding to the highest €pycno values » this is not visible on current figure 3.

We think Fig 3d mirrors 3c spikes quite clearly on Fig 3, but take reviewer 2's comment on board and have marked some of the oxygen flux spikes out with grey boxes.

- L.142 to 144 : add figure references to help the reading

Noted, we have updated the manuscript to reference the figures more frequently.

- L.145 – 151 : see 'spike' comments « general » section

We have added a section here describing our rationale for choosing this value of epsilon you describe as 'spike'.

Conclusion includes a discussion part that is missing in the paper. Change the section title or include a section « discussion »

In response to both Reviewers 1 and 2, we have amended the manuscript to include a discussion section and a shorter conclusions section.

Technical corrections

- L. 57 – 61: add the time coverage of your dataset in this section compare to 17 month of data for Alter ECO for clarification. For example, <(Sep. to Nov.)> just before « autumn breakdown of stratification in the North Sea » and/or < Using this 3-month dataset >

Thank you; we have clarified this in L72 -L73.

- L70 : replace <('Kelvin', unit 444)> by <(called « Kelvin », unit 444)>

Thank you; we have changed this.

- L75 : replace <Kelvin was equipped … > by <Glider « Kelvin » was equipped … > as line 113

We have corrected this in the text.

- L118 : add model references.

This is in comparison to AMM7 temperature, which we have added as Appendix C and referred to in the manuscript main text (L129).

[Figure]

Fig. C1 shows temperatures (deg C) from AMM7 for the area where the glider was sampling, stratification shown to break down by October 2018.

- L142 : replace <-34-4> by <34.4>

Changed

- L150 1.0 mmol m-2 s-1 for A and 1 mmol m-2 s-1 for B > harmonize

Have changed thank you (L159).

- figure 3  caption : add <(black line)> after <within the pycnocline>

Changed

- Explain TKE abbreviation in the text

Done (L130).

- Figure improvement suggestions : add bottom depth of the shelf sea on the sections of Figure 2 and 3 to complete them

We agree and have included the bottom depth to improve the figures.